# The Potential Role of Dysregulated miRNAs in Adolescent Idiopathic Scoliosis and 22q11.2 Deletion Syndrome

**DOI:** 10.3390/jpm12111925

**Published:** 2022-11-18

**Authors:** Nicola Montemurro, Luca Ricciardi, Alba Scerrati, Giorgio Ippolito, Giorgio Lofrese, Sokol Trungu, Andrea Stoccoro

**Affiliations:** 1Department of Neurosurgery, Azienda Ospedaliera Universitaria Pisana (AOUP), University of Pisa, 56100 Pisa, Italy; 2Department of NESMOS, Sapienza University of Rome, 00185 Roma, Italy; 3Department of Translational Medicine, University of Ferrara, 44121 Ferrara, Italy; 4Istituto Chirurgico Ortopedico Traumatologico (ICOT), DSBMC Sapienza Università di Roma-Polo Pontino, 04100 Latina, Italy; 5Division of Neurosurgery, Ospedale Bufalini, 47023 Cesena, Italy; 6Department of Translational Research and of New Surgical and Medical Technologies, University of Pisa, 56100 Pisa, Italy

**Keywords:** scoliosis, 22q11.2 deletion syndrome, DiGeorge syndrome, adolescent idiopathic scoliosis, microRNA

## Abstract

*Background:* Adolescent idiopathic scoliosis (AIS), affecting 2–4% of adolescents, is a multifactorial spinal disease. Interactions between genetic and environmental factors can influence disease onset through epigenetic mechanisms, including DNA methylation, histone modifications and miRNA expression. Recent evidence reported that, among all clinical features in individuals with 22q11.2 deletion syndrome (DS), scoliosis can occur with a higher incidence than in the general population. *Methods:* A PubMed and Ovid Medline search was performed for idiopathic scoliosis in the setting of 22q11.2DS and miRNA according to PRISMA guidelines. *Results:* Four papers, accounting for 2841 individuals, reported clinical data about scoliosis in individuals with 22q11.2DS, showing that approximately 35.1% of the individuals with 22q11.2DS developed scoliosis. *Conclusions:* 22q11.2DS could be used as a model for the study of AIS. The *DGCR8* gene seems to be essential for microRNA biogenesis, which is why we propose that a possible common pathological mechanism between scoliosis and 22q11.2DS could be the dysregulation of microRNA expression. In the current study, we identified two miRNAs that were altered in both 22q11.2DS and AIS, miR-93 and miR-1306, thus, corroborating the hypothesis that the two diseases share common molecular alterations.

## 1. Introduction

Scoliosis, a three-dimensional (3D) rotational deformity of the spine and trunk, is defined as a lateral deviation of the spine of at least a ten degree Cobb angle [1]. Adolescent idiopathic scoliosis (AIS) affects approximately 2–4% of adolescents around the world [2,3]. The etiology of AIS is multifactorial, involving both genetic and environmental factors, such as metabolic, hormonal and biomechanical factors [3,4]. The early diagnosis and accurate prediction of curve progression are very important in AIS, as they can help clinicians to avoid the negative effects of AIS treatments including brace therapy, scoliosis surgery and frequent exposure to radiation. Therefore, the identification of biomarkers that could help in the diagnosis of individuals in the early stages of the disease is of the utmost importance, as well as in the prognosis, which could direct clinicians to the identification of the best treatment for the individual. A major problem in the research of AIS is that most individuals come to receive medical attention after the manifestation of the curve; thus, it becomes very difficult to investigate these biomarkers before the disease manifests and in the very early stages of the disease [5].

Recent evidence reported that individuals with 22q11.2 deletion syndrome (22q11.2DS), also known as DiGeorge syndrome or velocardiofacial syndrome, are more affected by AIS compared to the general population, usually develop scoliosis in adolescence during a growth spurt and the majority of individuals have an idiopathic-like curve pattern [6]. For these reasons, it has been proposed to use this syndrome as a model for the study of AIS etiology [6,7]. 22q11.2DS occurs in approximately 1 in 4000–6000 newborns, making it the most common microdeletion disorder in humans. 22q11.2DS is associated with a spectrum of variable clinical phenotypes, including congenital heart disease (CHD), palatal abnormalities, laryngotracheoesophageal and gastrointestinal anomalies, immune deficiency, but also skeletal anomalies, including occipital–cervical malformations, scoliosis, rib and vertebral anomalies, clubfoot and polydactyly [8]. The genes within the 22q11.2 deleted region encode for proteins involved in several cellular pathways, including chromatin modifications, cell signaling, cell–cell adhesion, protein trafficking, mRNA/miRNA biogenesis, gene transcription, transmembrane receptors/transporters, mitochondrial metabolism and homeostasis [9]. Recent research has uncovered novel genetic variants, such as single-nucleotide polymorphisms (SNPs) and copy number variations (CNVs), and epigenetic differences in 22q11.2DS individuals that could influence disease severity. Among the epigenetic mechanisms involved in 22q11.2DS, of particular interest are the microRNAs (miRNAs), given that a gene included in the chromosomal region deleted in the 22q11.2DS, the *DGCR8* gene, plays a primary role in miRNA biogenesis [10]. The *DGCR8* gene encodes for a nuclear miRNA binding protein required for the biogenesis of miRNAs, and, as expected, the haploinsufficiency of *DGCR8* interferes with the processing of miRNAs, and, as such it is emerging that some of the phenotypic features of 22q11.2DS can be ascribed not only to the haploinsufficiency of genes within the deleted region, but also to an altered global dosage of miRNAs [11].

Interestingly, recent evidence suggests that altered miRNA expression could also be involved in the etiology of AIS. In fact, miRNAs have been proposed as important contributors in bone morphogenesis and osteoclastogenesis, which makes them interesting biomolecules to study the molecular causes of scoliosis [12]. Moreover, altered miRNA expression has been reported in specimens from individuals with AIS [13]. Given that 22q11.2DS has been proposed as a model to study molecular pathological mechanisms underlying AIS, and that altered miRNA expression seems to be a common feature in both 22q11.2DS and AIS, it could be hypothesized that the identification of commonly altered miRNA expressions in these diseases could help to better understand AIS etiopathogenesis, potentially providing diagnostic or prognostic biomarkers for this disease. To the best of our knowledge, no previous papers reported on research for miRNA expression profiles in 22q11.2DS and AIS individuals. In order to validate this hypothesis, we performed a review of the literature, searching for correlations between scoliosis and 22q11.2DS, between miRNA dysregulation and 22q11.2 and between miRNA dysregulation and scoliosis.

## 2. Materials and Methods

A PubMed and Ovid Medline search was performed for idiopathic scoliosis in the setting of 22q11.2DS and miRNA. PRISMA guidelines (Preferred Reporting Items for Systematic Reviews and Meta-Analyses) were followed [14]. The key words “22q11.2 deletion syndrome”, “DiGeorge syndrome”, “Velo-cardio-facial syndrome”, “microRNA”, “miRNA” and “idiopathic scoliosis” were used in both “AND” and “OR” combinations. The inclusion criteria were as follows: papers reporting on data about 22q11.2DS individuals with scoliosis and papers reporting on data about 22q11.2DS or scoliosis with miRNA. The exclusion criteria were as follows: (1) review articles, (2) studies published in languages other than English, (3) studies with animal subjects and (4) studies not related to the topic. Using the “PubMed Advanced Search Builder” and “Ovid’s Advanced Search Mode” with the three queries “(22q11.2DS) AND (idiopathic scoliosis) AND (miRNA)” producing no results. In order to obtain some findings, we conducted the three-query research as follows “((22q11.2DS) AND (idiopathic scoliosis)) OR ((22q11.2DS) AND (miRNA)) OR ((idiopathic scoliosis) AND (miRNA))”, or, for convenience, three different queries that produced interesting results as follows: “(22q11.2DS) AND (idiopathic scoliosis)”, “(22q11.2DS) AND (miRNA)” and “(idiopathic scoliosis) AND (miRNA)”. All this complied with the PRISMA guidelines.

### Quality Scoring

A modified version of the Newcastle–Ottawa scale (NOS) was used for the quality assessment of the included studies [15]. Two authors (N.M. and A.S.) performed the quality assessment independently, and any disagreement between them was resolved by a third author (L.R.). All studies included in this review were rated with an NOS ≥ 5.

## 3. Results

The database search for 22q11.2DS and idiopathic scoliosis yielded 87 articles (Figure 1). After the removal of duplicates, 40 articles were eligible for screening. A total of 28 articles met the selection criteria as described above, according to the PRISMA guidelines [5,6,16,17,18,19,20,21,22,23,24,25,26,27,28,29,30,31,32,33,34,35,36,37,38,39,40,41]. Four papers, accounting for 2841 individuals, reported clinical data about scoliosis in individuals with 22q11.2DS, showing that approximately 35.1% of the individuals with 22q11.2DS developed scoliosis, whereas 64.3% of individuals had CHD, which represented the most common clinical presentation in all studies reported [18,25,26,32]. Studies reported that the occurrence of scoliosis was not associated with the presence of CHD [25]. Homans et al. [25] reported that 20.4% of 1085 individuals with 22q11.2DS had scoliosis, but this prevalence reached 47.9% when considering individuals older than 16 years old. Sixty-three percent of all individuals with scoliosis had a scoliotic curve pattern that resembled AIS [25]. Table 1 shows all details.

The database search for 22q11.2DS and miRNA yielded 174 articles (Figure 2). After the removal of duplicates, 99 articles were eligible for screening. A total of four articles (Table 2) met the selection criteria as described above according to the PRISMA guidelines [11,42,43,44]. These papers revealed that several miRNAs were dysregulated in biological specimens from 22q11.2DS individuals. Collectively, the selected four papers identified alterations of miRNAs that were involved in several pathways, including neurological, immune system and cardiovascular pathways, as well as in embryonic and skeletal development. A few miRNAs were replicated from at least two different papers. Three articles [42,43,44] reported the deregulation of miR-185, while two articles [43,44] reported the deregulation of miR-150, miR-194 and miR-363.

The database search for idiopathic scoliosis and miRNA yielded 45 articles (Figure 3). After the removal of duplicates, 26 articles were eligible for screening. Ten papers that investigated miRNAs in specimens from AIS individuals focused on the mechanisms underlying intervertebral disc degeneration, so they were not included in the current review given that they used AIS samples as control samples [16,29,30,38,39,41,45,46,47,48]. A total of six articles (Table 3) met the inclusion criteria and was included in the present review [23,33,40,48,49]. Collectively, the six selected papers identified the alterations of miRNAs involved in apoptosis, cell adhesion, transmembrane transport, immune responses, as well as in the regulation of the bone morphogenic protein and Wnt/β-catenin pathway, which are important in osteoblast/osteoclast differentiation and bone metabolism. Two papers [45,49] identified the alterations of miR-15a in the inferior facet of joint cartilage and in the bone marrow mesenchymal stem cells of AIS individuals. By comparing dysregulated miRNAs in 22q11.2DS and AIS, two miRNAs, miR-93 and miR-1306, were found to be dysregulated in both diseases (Figure 4).

## 4. Discussion

### 4.1. Clinical Multifactorial Analysis

In recent years, a primary muscle disorder linked with genetic mutations was postulated as a possible etiology of idiopathic scoliosis [51]. Recently, 22q11.2DS was proposed to be used as a model for AIS and that within 22q11.2DS, the causal pathways leading to scoliosis could be identified, which should be used for the general population [6]. Our review found that scoliosis occurred in 35.1% of individuals with laboratory-confirmed 22q11.2DS. The scoliosis had clinical characteristics of AIS. The evidence for the high incidence of scoliosis within 22q11.2DS was limited. To the best of our knowledge, only four studies reported on this in the literature. Scoliosis in individuals with 22q11.2DS represents an important spinal deformity, with surgical intervention applying to 5–6.4% of all 22q11.2DS individuals [31,52]. Other spinal deformities, such as upper cervical anomalies, spina bifida and sacral myelomeningocele, were reported in these individuals [53,54,55].

### 4.2. MicroRNA and 22q11.2DS

We found four papers in which miRNA expression or miRNA-coding genes were investigated in individuals with 22q11.2DS. In one of these papers, copy number variations (CNVs) were investigated in the peripheral blood cells of 21 individuals with 22q11.2DS, and 11 CNVs in which mapped miRNA-codifying genes were detected in 19 individuals [11]. Some of these CNVs in the miRNAs were deleted, some duplicated, and some others were present as both deletions and duplications; it was interesting to note that all those miRNAs had a biological role in pathways related to some of the phenotypic characteristics of 22q11.2DS. Merico and colleagues [42], by using computational tools, detected seven miRNAs encoded within the typical 22q11.2 deleted region, which are involved in neuronal processes and in developmental networks, as well as in the bone morphogenic protein (BMP) group of growth factors and SMAD and transforming factor beta (TGF-beta) signaling.

Sellier et al. [43] and De la Morena et al. [44] investigated the expression of several miRNAs in the peripheral blood of AIS individuals and control subjects, finding alterations in miRNAs that are involved in neurological, immune system and cardiovascular pathways. Overall, these four papers revealed a potential impairment in different miRNAs in the etiology of 22q11.2DS. Interestingly, three articles reported on the deregulation of miR-185 [42,43,44], and two articles reported on the deregulation of miR-150, miR-194 and miR-363 [43,44]. The observation of the dysregulation of miR-185 was not surprising, given that the gene that encodes for this miRNA is located within the deleted region of chromosome 22. Indeed, all three articles reported a strong downregulation of miR-185 expression. Merico et al. [42] reported that miR-185, together with another three miRNAs that mapped the 22q11.2 deleted region, miR-1286, miR-3618 and miR-4761, was involved in embryonic development, in the mitogen-activated protein kinase cascade, in the bone morphogenetic protein group of growth factors and in SMAD and transforming growth factor beta signaling, all pathways that were found to be involved in scoliosis [55,56].

The potential involvement of miRNA dysregulation in 22q11.2DS is well known, given that among the genes mapped in the 22q11.2 deleted region was the *DGCR8* gene, which encodes a crucial component of the microprocessor complex that contributes to miRNA biogenesis and, therefore, to global gene regulation. Thus, miRNA dysregulation in 22q11.2DS could be due to both the altered expression of miRNAs mapped in the 22q11.2 deleted region and the miRNAs being dysregulated due to the haploinsufficiency of the *DGCR8* gene.

### 4.3. MicroRNA and AIS

Several authors suggested that miRNA expression dysregulation could be an epigenetic mechanism that plays a role in the pathogenesis of scoliosis. In the current study, we found six papers that identified alterations in miRNA expression or in miRNA-encoding genes in individuals with idiopathic scoliosis [23,33,40,45,49,50,57,58]. Li et al. [45] reported that miR-15a, which targets Bcl2, was downregulated in the spinal facet joint cartilage of individuals with scoliosis. Hui et al. [49] investigated the expression of several miRNAs in bone marrow mesenchymal stem cells from AIS and control individuals, and identified 54 differentially expressed miRNAs in the specimens from AIS individuals. Interestingly, among them, miR-15a was found to be dysregulated in the paper by Li et al. [58]. Jiang et al. [57] investigated transcriptomic differences between the two sides of the paravertebral muscle and demonstrated that the mRNA expression of two genes, the adiponectin, C1Q and collagen domain containing (*ADIPOQ*) gene and the H19-imprinted maternally expressed transcript (*H19*) gene, was differentially expressed between the two sides of paravertebral muscle, and a lower expression of *H19* and higher expression of *ADIPOQ* mRNA in the muscle were positively correlated with curve severity and age at initiation. The authors observed that muscle tissue expression levels of miR-675-5p, encoded by *H19*, were positively correlated with that of H19 and negatively with that of *ADIPOQ*. By means of bioinformatics algorithms and in vitro investigations, it was observed that miR-675-5p was able to bind the 3′UTR of the *ADIPOQ* gene, leading to a reduced expression of the ADIPOQ protein, further suggesting that miRNAs could play a role in the regulation of genes involved in the onset and progression of scoliosis [57]. Zhang et al. [40] found that miR-145 was overexpressed in bone tissues and primary osteoblasts in individuals with scoliosis, and that serum miR-145 was negatively correlated with bone markers, including sclerostin, osteopontin and osteoprotegerin. García-Giménez and colleagues [23] reported a differential expression of several miRNAs in the peripheral blood of individuals with scoliosis compared to control subjects, suggesting that these miRNAs could participate in scoliosis pathogenesis by regulating osteoblast and osteoclast activity. Finally, by means of a genome-wide association investigation performed in 2543 AIS subjects, a functional variant in MIR4300HG, the host gene of the miRNA MIR4300, was found to be associated with AIS curve progression [27]. Further evidence of the important role of miRNAs in AIS etiopathogenesis was derived from an animal study in which rat embryos (gestation day 9) exposed to air pollution exhibited a differential expression of 291 miRNAs compared to the non-exposed group, and, by using bioinformatic analyses, the authors predicted that the dysregulated miRNAs play crucial roles in the pathogenesis of congenital spinal defects by deregulating multiple biological processes [59].

### 4.4. Unifying miRNA Dysregulation Observations in 22q11DS and AIS

By comparing dysregulated miRNAs in 22q11.2DS and in AIS, two miRNAs, miR-93 and miR-1306, were found to be dysregulated in both diseases (Table 2 and Table 3). Particularly, miR-93 was found to be downregulated by De la Morena and coworkers, and upregulated by Hui et al. [49]. This difference could be due to different specimens used for the analyses. In fact, De la Morena and coworkers investigated miRNA expression in the peripheral blood of 22q11.DS individuals, while Hui et al. [49] used bone marrow aspirates from AIS individuals. However, it was interesting that miR-93 was dysregulated in both 22q11DS and AIS specimens, given that this miRNA is involved in osteogenic differentiation by targeting the bone morphogenetic protein-2, in osteoblast mineralization and in the bone mineral density, which are well-established pathways in the etiology of scoliosis [60,61,62,63,64,65,66].

Regarding miR-1306, it was found to be downregulated in the peripheral blood of AIS individuals, and Merico and coworkers reported that the gene encoding miR-1306 was located in the 22q11.2 deleted region, meaning that it was also downregulated in 22q11.2DS individuals [23,42]. Particularly, miR-1306 was encoded in the genomic sequence of the *DGCR8* gene [42]. García-Giménez and collaborators found that miR-1306-3p targeted the protein serine/threonine phosphatase 2CB, which participates in the TGFβ signaling pathway, which has already been shown to be involved in AIS, as an increased expression of *TGF-β2*, *TGF-β3* and *TGFBR2* (encoding for a TGFβ receptor) was found in samples from the curve concavity of AIS patients [67] and it was reported that *TGFB1* may play a role in the curve progression of AIS [68]. Moreover, miR-1306-3p also targets Rac2, which is an essential Rho GTPase in mature osteoclasts for chemotaxis and resorptive activity, while among the target miR-1306-5p. there was bone morphogenic protein 1, which is involved in bone and cartilage development [69,70]. Therefore, the altered expression of miR-93 and miR-1306 could be involved in the pathological mechanism underlying scoliosis in both 22q11.2DS and AIS.

Further suggestions for a common miRNA expression alteration link between 22q11DS and AIS were derived from observations of the involvement of *DGCR8* in bone turnover, the impairment of which is strongly involved in scoliosis [71,72,73]. Sugatani and colleagues [71] observed that the osteoclast-specific deletion of *DGCR8* in mice led to impaired osteoclastic development and bone resorption, so that bone development was significantly retarded. In cell cultures, the expression levels of osteoclastic phenotype-related genes and proteins were remarkably inhibited during osteoclastogenesis in *DGCR8* deficiency, evidencing that *DGCR8*-dependent miRNAs were indispensable for the osteoclastic control of bone metabolism [71]. Similarly, Choi et al. [72] generated mice, in which *DGCR8* was conditionally deleted in osteoprogenitor cells, observing alterations in osteoblast differentiation. These observations provide evidence that miRNA dysregulation underlies pathological mechanisms in both 22q11.2DS and AIS, and that they share dysregulations of the same miRNAs (Figure 5).

## 5. Conclusions

In conclusion, we confirmed that scoliosis is a frequent hallmark of individuals with 22q11.2DS and that this syndrome could be a useful model for the discovery of the pathological mechanisms underlying AIS. In particular, a miRNA alteration investigation in 22q11.2DS could provide new insight in the etiopathogenesis of AIS. In the current study, we identified two miRNAs that were altered in both 22q11.2DS and AIS, miR-93 and miR-1306, thus, corroborating the hypothesis that the two diseases share common molecular alterations, although it is likely that there are many others that have yet to be discovered. Further investigations are needed to identify other potential miRNAs involved in the genesis of scoliosis in the two diseases, as well as to confirm the connection between 22q11.2 and scoliosis pathological mechanisms. Since 22q11.2DS is a congenital disability that can be diagnosed during infancy, and scoliosis affects at least 35% of these individuals, this syndrome offers the opportunity to identify early biomarkers for scoliosis development and severity valid also for AIS. For example, miRNA expression investigations in 22q11.2 individuals with and without scoliosis, as well as in individuals with AIS, could be useful for identifying specific biomarkers of scoliosis and, subsequently, for expanding these findings to the general population to clarify which miRNAs are directly involved in scoliosis onset and development. Moreover, scoliosis monitoring should be part of the clinical management of individuals with 22q11.2DS [32]. The identification of dysregulated miRNAs specific to AIS could help with the development of early diagnostic biomarkers of the disease, which could allow earlier postural interventions and could provide new targets for the development of pharmacological therapies. More importantly, future studies should not only be aimed at looking for possible miRNAs associated with scoliosis, but also at its severity, in order to identify not only biomarkers for disease diagnosis, but also for disease prognosis.

## Figures and Tables

**Figure 1 jpm-12-01925-f001:**
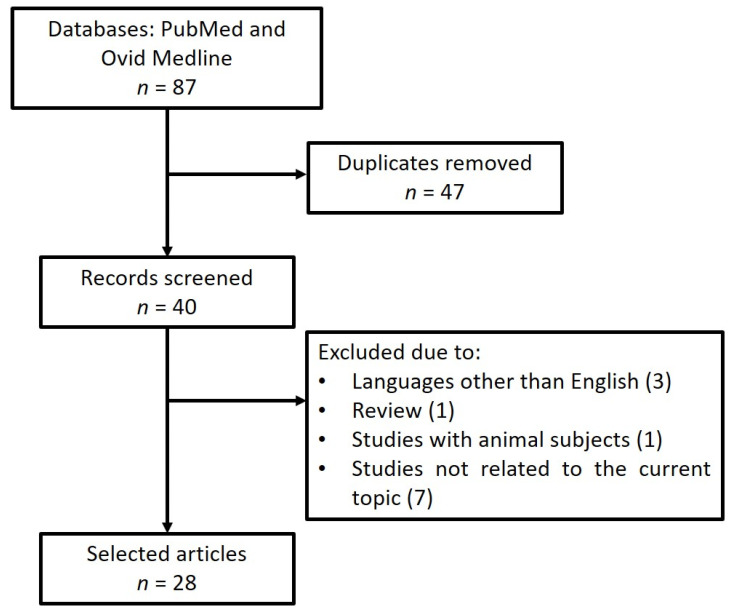
PRISMA flow diagram of study selection for 22q11.2 deletion syndrome and scoliosis.

**Figure 2 jpm-12-01925-f002:**
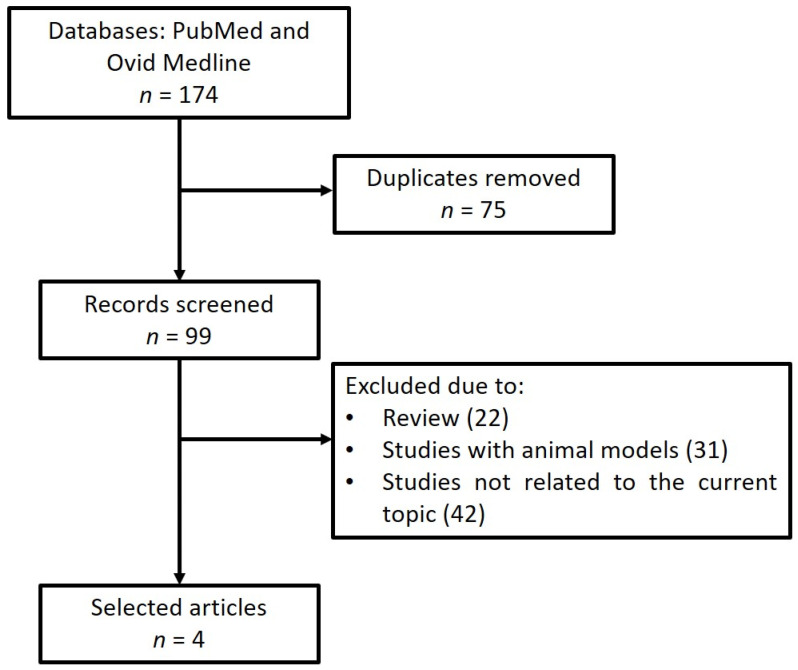
PRISMA flow diagram of study selection for 22q11.2 deletion syndrome and miRNA dysregulation.

**Figure 3 jpm-12-01925-f003:**
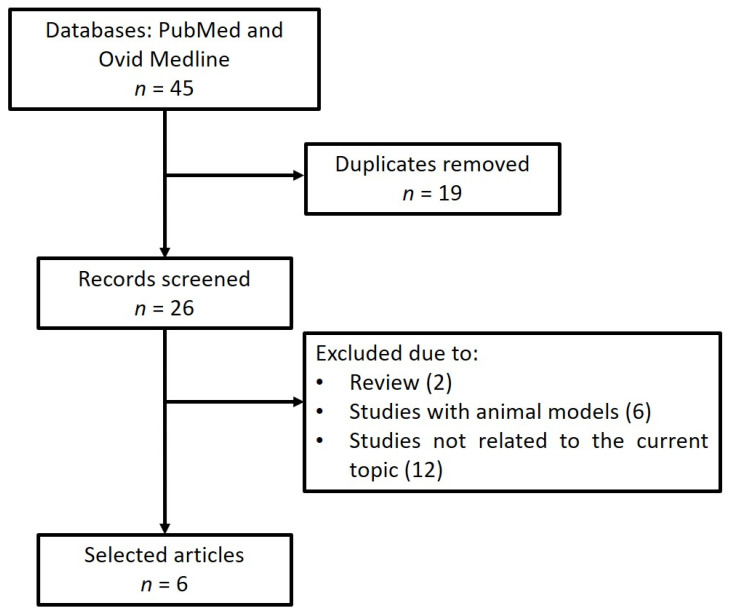
PRISMA flow diagram of study selection for idiopathic scoliosis and miRNA dysregulation.

**Figure 4 jpm-12-01925-f004:**
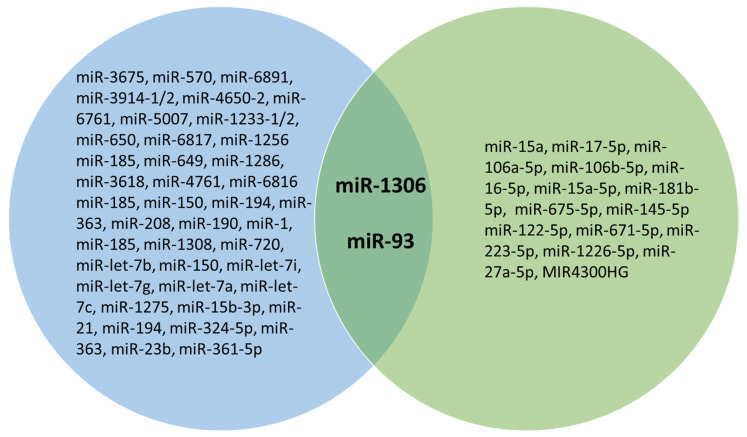
Venn diagram showing miRNAs dysregulated in 22q11.2DS (blue circle) and in adolescent idiopathic scoliosis (green circle). The two miRNAs dysregulated in both diseases are in bold.

**Figure 5 jpm-12-01925-f005:**
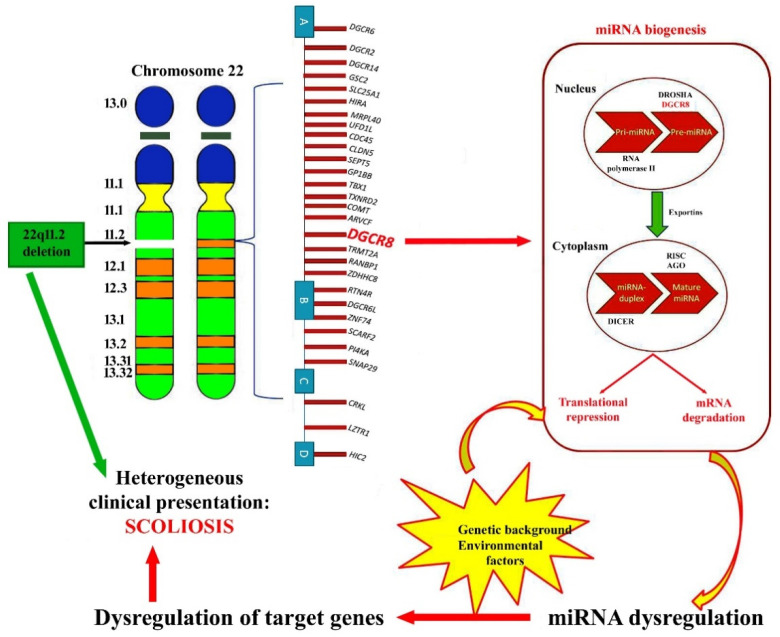
Schematic representation of the potential role of dysregulated miRNAs in 22q11.2DS and scoliosis. Individuals usually harbor one of three common deletion sizes (3, 2 and 1.5 Mb) between low-copy repeats (LCRs) designated as A-D, A-C and A-B, respectively, which leads to haploinsufficiency of several genes, including *DGCR8*, which is involved in miRNA biogenesis. MiRNA binds to mRNA transcripts, targeting them for degradation, thus, playing a pivotal role in regulating mRNA translation. The genetic background of individual and environmental factors could interfere with miRNA expression, leading to dysregulation of their target genes.

**Table 1 jpm-12-01925-t001:** Analysis of the current literature for individuals with 22q11.2 deletion syndrome and scoliosis.

	Year	Individuals (*n*)	Mean Age	Sex (M/F)	Scoliosis (%)
Morava et al. [32]	2002	20	na	na	3 (15)
Homans et al. [25]	2018	1085	11.4	560/525	221 (20.4)
Campbell et al. [18]	2018	1421	na	725/696	710 (50)
Homans et al. [26]	2020	315	23	na	64 (20.3)
		2841	-	-	998 (35.1)

na, not available.

**Table 2 jpm-12-01925-t002:** Selected papers about 22q11.2 deletion syndrome and alteration of microRNA.

Experimental Model	Method of Detection	Affected miRNAs	Description of Main Findings	Reference
Array CGH in peripheral blood of 21 individuals with 22q11.2DS	Array CGH	miR-3675	In 19 individuals, CNVs containing miRNAs were identified, suggesting an altered dosage of those miRNAs.	Bertini et al. [11]
		miR-570		
		miR-6891		
		miR-3914-1/2		
		miR-4650-2		
		miR-6761		
		miR-5007		
		miR-1233-1/2		
		miR-650		
		miR-6817		
		miR-1256		
Investigation of miRNAs overlapped with 22q11.2 microdeletion and investigation of predicted target genes	Genome database analysis	miR-185	Authors identified seven miRNAs encoded within the typical 22q11.2 deleted region. The 22q11.2 deletion region was characterized by high miRNA density. Functional enrichment profiles of the 22q11.2 region miRNA target genes suggested a role in neuronal processes and broader developmental networks.	Merico et al. [42]
		miR-649		
		miR-1286		
		miR-1306		
		miR-3618		
		miR-4761		
		miR-6816		
Selected miRNA expression analyses in peripheral blood of 30 individuals with 22q11.2DS and 15 controls	Quantitative PCR	miR-185	MiR-185, miR-150, miR-194 and miR-363 were downregulated in individuals with 22q11.2DS, as compared to TD, while miR-208, miR-190 and miR-1 were upregulated. Authors also reported decreased expression levels of genes within the deletion region of chromosome 22, including *DGCR8*.	Sellier et al. [43]
		miR-150		
		miR-194		
		miR-363		
		miR-208		
		miR-190		
		miR-1		
Selected miRNA expression patterns in the peripheral blood of individuals with 22q11.2DS (*n* = 31) and normal controls (*n* = 22)	MicroRNA array expression profiling (containing over 600 human miRNA probes)	miR-185	Eighteen miRNAs were differentially expressed in a statistically significant manner between 22q11.2DS individuals and controls.	De la Morena et al. [44]
		miR-1308		
		miR-720		
		miR-let-7b		
		miR-150		
		miR-let-7i		
		miR-93		
		miR-let-7g		
		miR-let-7a		
		miR-let-7c		
		miR-1275		
		miR-15b-3p		
		miR-21		
		miR-194		
		miR-324-5p		
		miR-363		
		miR-23b		
		miR-361-5p		

CNVs, copy number variations; TD, typical developing control subject.

**Table 3 jpm-12-01925-t003:** Selected papers regarding idiopathic scoliosis and alteration of microRNAs.

Experimental Model	Method of Detection	Affected miRNAs	Description of Main Findings	Reference
Investigation of several miRNAs, including miR-15a, miR-16, miR-let7d and miR-29b in the inferior facet joint cartilage in 11 AIS individuals and 10 controls	Real-time PCR	miR-15a	Expression of miR-15a was downregulated in IS individuals, dysregulating the miR-15a/Bcl2 pathway, thus, affecting the proliferation of chondrocytes and leading to abnormal spine growth, which resulted in the development and progression of IS	Li et al. [45]
Investigation of miRNA expression in bone marrow aspirates from 40 AIS individuals and 25 non-AIS individuals	MiRNA microarray and real-time PCR	miR-17-5p	The study identified 54 differentially expressed miRNAs in bone marrow mesenchymal stem cells from AIS individuals, and interaction network analysis indicated that 7 most significant central miRNAs may play essential roles in AIS pathogenesis and accompanied osteopenia.	Hui et al. [49]
		miR-106a-5p		
		miR-106b-5p		
		miR-16-5p		
		miR-93-5p		
		miR-15a-5p		
		miR-181b-5p		
RNA sequencing in five pairs of paravertebral muscles from five AIS individuals	RNA sequencing and real-time PCR	miR-675-5p	*ADIPOQ* and *H19* genes were differentially expressed between two sides of paravertebral muscles and were associated with larger spinal curves and earlier age at initiation. *ADIPOQ* gene was regulated by miR-675-5p, which is encoded by *H19*.	Jiang et al. [50]
Bone biopsies from 13 individuals with AIS and 10 control subjects and the primary osteoblasts derived from those samples were used to identify the potential for osteoblast and osteocyte functions to interfere with miRNA candidates	MiRNA microarray and real-time PCR	miR-145-5p	Microarray analysis identified overexpression of miRNA-145-5p in bone biopsies of AIS individuals. On the other hand, the difference in plasma levels of miR-145 was not statistically significant between the control and AIS groups. However, significant negative correlations between circulating miR-145 and serum sclerostin, osteopontin and osteoprotegerin were observed in individuals with AIS but not in controls.	Zhang et al. [40]
Peripheral blood of 30 AIS individuals and 13 healthy controls	NGS and real-time PCR	miR-122-5p,	NGS analyses showed that miR-671-5p and miR-1306-3p were under-represented, while miR-1226-5p and miR-27a-5p were present at high levels in the plasma of the individuals compared to the controls. miR-223-5p and miR-122-5p were homogenously over-represented in the individuals, but their expression was heterogeneous among the controls. RT-PCR detected all the miRNAs derived from the previous NGS study, except for miR-1226-5p.	García-Giménez et al. [23]
		miR-671-5p,		
		miR-223-5p,		
		miR-1226-5p,		
		miR-27a-5p		
		miR-1306-3p		
Genome-wide association study in 1937 individuals with AIS divided into progression (*n* = 1105) and nonprogression groups (*n* = 832)	GWAS	*MIR4300HG*	GWAS analysis identified a functional SNP associated with AIS curve progression. This SNP was found in a putative enhancer region in intron 1 of MIR4300HG, and its risk allele showed significantly lower enhancer activities than the nonrisk allele. These findings indicated that MIR4300HG is the gene responsible for AIS curve progression in the locus and its decreased expression would lead to a curve progression.	Ogura et al. [33]

GWAS, genome-wide association; NGS, next-generation sequencing; PCR, polymerase chain reaction.

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
