# Peer review of "The Potential Role of Dysregulated miRNAs in Adolescent Idiopathic Scoliosis and 22q11.2 Deletion Syndrome"

_jpm, 2022, doi:10.3390/jpm12111925_

Round 1

Reviewer 1 Report (New Reviewer)

Please see attached comments.

Author Response

We appreciated the reviewer's comments as it gives us the opportunity to improve our manuscript.

  • we changed the word "patients" with "individuals"
  • we revised that sentence
  • we removed “This is a table”
  • we added abbreviations in tables 2 and 3
  • Grammar: “are” instead of “is”. Revised.
  • We revised that paragraph and the related citations.
  • We added citation [49]
  • We completed sentence at lines 298-300
  • “share” instead of “shared”. Revised at line 315.

Reviewer 2 Report (New Reviewer)

The authors reviewed publications related to role of miRNA dysregulation in scoliosis.

It is a systemic review paper. The authors started with the observation that scoliosis was a common phenotype in 22q11.2 deletion syndrome (also known as DiGeorge syndrome). 

If they started with this condition, they should focus (at least with some section) (1) to show if there are common miRNA dysregulation in  22q11.2 deletion syndrome and scoliosis of other etiology (idiopathic scoliosis).

(2) Is there any common miRNA dysregulation among studies that reviewed ?

(3) show such  results in a Venn diagram 

(4) if they could identify one such common miRNA, go deep into the possible mechanisms that it causes scoliosis

(5) if no such common miRNA is found, please provide explanation. 

Author Response

We appreciated the reviewer's comments as it gives us the opportunity to improve our manuscript.

  1. The section “4.4. Unifying miRNAs dysregulation observations in 22q11DS and AIS.” has now been implemented with further information regarding the common miRNAs dysregulated in 22q11.DS and adolescent idiopathic scoliosis.
  2. The two common miRNAs dysregulated in 22q11.2DS and AIS, miR-93 and miR-1306, have been now clearly shown in the Venn diagram, and their potential involvement in the pathogenesis of scoliosis further discussed in the section 4.4.
  3. Following the reviewer’s suggestion, the Venn diagram showing the altered miRNAs in 22q11.2DS and AIS has been added in the revised version of the manuscript (see new figure 4)
  4. The possible mechanisms linked to scoliosis driven by the same miRNAs altered in both 22q11.2DS and AIS have been included in the section “4.4. Unifying miRNAs dysregulation observations in 22q11DS and AIS”.
  5. Please, see reply to point 4.

This manuscript is a resubmission of an earlier submission. The following is a list of the peer review reports and author responses from that submission.

Round 1

Reviewer 1 Report

1.      First of all, most AIS show itself as a mild curvature, how to apply the “ hypothetic result” to explain who will progress to severe curvature or not by only two “miRNAs”?

2.      Second, since 22q11.2DS is congenital disease, does the scoliosis occur since 22q11.2DS patient from infant, instead of from “adolescent”? If that true, how to compare your hypothesis model “22q11.2DS and AIS share the same pathogenesis”?

3.      I concern the methodology, since it is a “systemic review and meta-analysis”, why the author put 2 different diseases to conclude the similar “hypothetic” finding? Instead of the “evidence-based” finding?

4.      Moreover, 3 PRISMA flow-diagrams make me confused. This is not regular PRISMA method in systemic review.

5.      In addition, the majority of results are summary of literature, could the author show “risk of bias assessment”, “forest plots of outcome” in regular method of systemic review.

Author Response

We appreciated the reviewer's comments as it gives us the opportunity to deepen the discussion on our hypothesis, as well as to make it clearer.

  1. We would like to clarify that we do not think that the use of the two miRNAs identified so far in common between the two diseases is sufficient to identify the subjects who will develop a severe curvature among AIS patients. As we reported in the “conclusions” section “In the current study we identified two miRNAs that are altered in both 22q11.2DS and AIS, miR-93 and miR-1306, thus corroborating the hypothesis that the two diseases shared common molecular alterations”, but likely there are many other miRNAs altered in both diseases that have yet to be discovered. Further investigations should be performed in order to identify other potential miRNAs involved in the onset of scoliosis in the two diseases. More importantly, future studies should be aimed not only at looking for possible miRNAs associated with scoliosis, but also at its severity, in order to identify not only biomarkers for disease diagnosis, but also for disease prognosis. We have now added these considerations in the section “5. Conclusions” (Lines 323-342) following the reviewer’s comment.
  2. This is an important point raised by the reviewer. It has been reported that 2DS patients usually develop scoliosis in adolescence during the growth spurt, similarly to what happens in AIS (Cheng et al., Nat Rev Dis Primers. 2015; Homans et al., Arch Dis Child. 2019; Homans et al., Med Hypotheses. 2019). This is one of the reasons for which 22q11.2DS has been proposed to be a model for the study of idiopathic scoliosis, in addition to the fact that most patients with 22q11.2 have an idiopathic-like curve pattern. So, it should be postulated that within 22q11.2DS causal pathways resulting in scoliosis can be identified and that these pathways may also play a role in scoliosis that affect the general population. Considering that 22q11.2 is a congenital disease that can be diagnosed during infancy, and that, as confirmed in the current manuscript, at least 35% of the patients develop scoliosis (unlike AIS which affects 2-4% of the general population), we have a great opportunity to identify early biomarkers for scoliosis development. In line with this, by detecting miRNAs expression differences between the 22q11.2 patients with and without scoliosis, we can identify specific biomarkers of scoliosis and subsequently expand these findings to the general population. Moreover, identification of miRNAs altered in both the conditions, could provide information regarding pathological mechanisms directly linked to scoliosis onset and development. We have now added these considerations in the introduction section (lines 50-60) and in the “conclusion” section (lines 323-342).
  3. The idea for the current manuscript derived by the hypothesis by Homans and collaborators (Homans et al., Med Hypotheses. 2019) that 22q11.2DS is a human disease that could be used as a model to investigate the pathogenic mechanisms underlying AIS. As a congenital disease, 22q11.2DS offers the opportunity to identify early pathological mechanism of scoliosis occurrence, as well as early biomarkers detectable before the onset of the scoliosis. Starting from this hypothesis, we propose that miRNAs dysregulation could be one of the pathogenetic mechanisms underlying both the two diseases. However, to the best of our knowledge, no investigation searched for common miRNAs expression profiles in 22q11.2DS and AIS patients. Consequently, it could not be possible to perform a systematic review on “evidence-based” findings, so that we performed two separate systematic review research to identify miRNAs dysregulated in 22q11.2DS patients and miRNAs dysregulated in AIS patients.
  4. Thank you for giving us the opportunity to explain our choice. We did a three steps PRISMA research as, also using “PubMed Advanced Search Builder” and “Ovid's Advanced Search Mode” with the three queries as follow "(22q11.2DS) AND (idiopathic scoliosis) AND (miRNA)" no results were found. In order to obtain some findings, what we technically did is three-query research with "PubMed Advanced Search Builder" as follow "((22q11.2DS) AND (idiopathic scoliosis)) OR ((22q11.2DS) AND (miRNA)) OR ((idiopathic scoliosis) AND (miRNA))". For convenience, we thought that readers can appreciate better this subdivision as we did in our manuscript with 3 different parts and 3 different PRISMA flow-diagram. We also reported this in Material and Methods section. Look at lines 98-105.
  5. It is not possible to perform “forest plots of outcome” for this type of review. However, we added a “2.1 Quality scoring” paragraph to we show what we did to reduce the risk of bias assessment. Look at lines 106-112.

We hope you would appreciate our revised version. Thank you for your time.

Reviewer 2 Report

The manuscript entitled “The Potential Role of Dysregulated miRNAs in Adolescent Idiopathic Scoliosis and 22q11.2 Deletion Syndrome” is well written and can be easily understood by readers in this field. This review paper has also explicitly stated its usefulness in explaining the relationship of clinical data about scoliosis in patients with 22q11.2DS (miR-93 and miR-1306). There are several notes for improvement:

1. This review using 3 combinations of keywords in parallel, namely: 22q11.2DS and idiopathic scoliosis obtained a total of 40 selected articles; 22q11.2DS and miRNA obtained 99 selected articles and; idiopathic scoliosis and miRNA obtained 26 screened articles. What is the purpose of this screening method? Why not do it in a step with using three keywords at once? The author must add the intended use of the method. Will it be aimed at elucidating the mechanism underlying the pathogenesis of idiopathic scoliosis?

2. Consistency in writing the search engine database. In the abstract and method it is written 'Pubmed and MEDLINE' while on the diagram it is written "Pubmed and OVID MEDLINE".

3. Column 2 the “miRNA predicted pathway” of table 2 is omitted because it indicates the authors' predictions and not data interpretations.

Author Response

We appreciated the reviewer's comments as it gives us the opportunity to improve our manuscript.

  1. Thank you for giving us the opportunity to explain our choice. We did a three steps PRISMA research as, also using "PubMed Advanced Search Builder" with the three queries as follow "(22q11.2DS) AND (idiopathic scoliosis) AND (miRNA)" no results were found. In order to obtain some findings, what we technically did is three-query research with "PubMed Advanced Search Builder" as follow "((22q11.2DS) AND (idiopathic scoliosis)) OR ((22q11.2DS) AND (miRNA)) OR ((idiopathic scoliosis) AND (miRNA))". For convenience, we thought that readers can appreciate better this subdivision as we did in our manuscript with 3 different parts and 3 different PRISMA flow-diagram. We also reported this in Material and Methods section. Look at lines 98-105.
  2. We changed it to "Ovid Medline" throughout the manuscript.
  3. We agree with the reviewer that the column “MiRNA predicted pathway” of table 2 could be misleading for the readers. We than removed that column from table 2 and table 3 following the reviewer’s suggestion. Look at lines 188-194.

We hope you would appreciate our revised version. Thank you for your time.

Round 2

Reviewer 1 Report

I am afraid it is not thought suitable for publication that due to the method and result are not formal systemic review and meta-analysis.